# Mannuronan C-5 Epimerases: Review of Activity Assays, Enzyme Characteristics, Structure, and Mechanism

**Zhongbin Xiao** [1,2], **Ming Sun** [2], **Tang Li** [2], **Miao Zhao** [1] and **Heng Yin** [2,*]

[1] Department of Materials and Chemicals, Dalian Polytechnic University, Dalian 116023, China
[2] Dalian Engineering Research Center for Carbohydrate Agricultural Preparations, Liaoning Provincial Key Laboratory of Carbohydrates, Dalian Institute of Chemical Physics, Chinese Academy of Sciences, Dalian 116023, China
[*] Correspondence: yinheng@dicp.ac.cn; Tel.: +86-(0411)-84379061

**Abstract:** Mannuronan C-5 epimerases (ManC5-Es) are produced by brown algae and some bacteria, such as Azotobacter and some Pseudomonas species. It can convert the transformation of β-D-mannuronic acid (M) to α-L-guluronic acid (G) in alginate with different patterns of epimerization. Alginate with different compositions and monomer sequences possess different properties and functions, which have been utilized in industries for various purposes. Therefore, ManC5-Es are key enzymes that are involved in the modifications of alginate for fuel, chemical, and industrial applications. Focusing on ManC5-Es, this review introduces and summarizes the methods of ManC5-Es activity assay especially the most widely used nuclear magnetic resonance spectroscopy method, characterization of the ManC5-Es from different origins especially the research progress of its enzymatic properties and product block distributions, and the catalytic mechanism of ManC5-E based on the resolved enzyme structures. Additionally, some potential future research directions are also outlooked.

**Keywords:** mannuronan C-5 epimerase; alginate; ManC5-Es characteristics; ManC5-Es structure





## 1. Introduction

Alginate is a type of natural polysaccharide mainly found in marine brown algae [1], including *Laminaria*, *Ascophyllum*, *Ecklonia*, *Lessonia*, *Durvillaea*, and *Macrocystisand* [1–3]. It is the most abundant component of the cell wall of brown algae [4,5]. Alginate is a linear β-1-4-linked polymer composed of β-D-mannuronic acid (M) and its C5 epimer α-L-guluronic acid (G) (Figure 1A) [6]. These two monomers are distributed continuously or alternately as blocks, forming three possible sequences: MM-blocks, GG-blocks, and MG-blocks (Figure 1A) [7]. The GG-blocks and MG-blocks [8] form gels with divalent ions such as $Ca^{2+}$ [9], whereas the MM-blocks can only form acid gels [10]. Due to the biocompatibility and gel strength of alginate, it has been applied in various fields including food, material, fuel, and chemical industries [11,12]. These physical properties of alginate are closely related to its compositions (M/G ratio, mainly ranging from 0.43 to 2.52 in typical species of brown seaweed), sequence, molecular weight (32,000–400,000 g/mol), and block distributions [13–16]. While the commercially available alginate is seaweed derived, of which the properties vary in the origin of brown algae species, harvest season, growth environment, and portion of the organisms [17], it requires an efficient way to modify the alginate to meet the industrial demands.

Some bacteria, such as *Azotobacter* species [18,19] and some *Pseudomonas* species [20,21], also produce alginate that could provide more defined structure and physical properties with the degradation, epimerization, and acetylation of alginate [11]. The biosynthesis pathway of alginate in bacteria involves a group of enzymes [19]. The mannuronan C5-epimerase (ManC5-E), which can convert M into G at the polymer level, can significantly alter the properties of alginate [22]. All alginate-producing organisms harbor *ManC5-Es*

genes [7], some of which, from bacteria and brown algae, have been subcloned with the encoded enzymes characterized [23,24].

In recent years, numerous alginate-modifying enzymes have been characterized, while most of them are focused on the alginate lyase [7]. With the development regarding the characteristics and mechanisms of ManC5-Es, it requires a timely review to summarize the progress of ManC5-Es in recent years. Thus, in this review, to better understand the ManC5-Es for further studies and applications, we summarized the methods of detecting epimerase activity, and the origins of ManC5-Es and their mechanism, as well as a prediction of the future research topics in this field.

**Figure 1.** The structure of alginate and the function of ManC5-Es. (**A**) Chemical structure of the repeating units of alginate. (**B**) The conversion of the β-D-mannuronate (M) to α-L-glucuronate (G) residues by ManC5-Es [7].

## 2. Methods for Analyzing of the Mannuronan C-5 Epimerases Reaction

ManC5-Es conduct the epimerization via β-elimination that follows the alginate lyase reaction [25], except for the glycosidic linkage breaking. Unlike alginate lyase that generates reducing ends or unsaturated saccharides after reaction, which can be simply detected by 3,5-dinitrosalicylic acid (DNS) or ultraviolet absorbance [26], the ManC5-Es go through a residue conversion of M to G, a transformation that cannot be easily detected. Therefore, sensitive methods for assaying and analyzing the epimerase activity or the mode of action are necessary and important to the ongoing research on algal and bacterial ManC5-Es. By far, several applicable methods for assaying epimerase activity in mannuronan C-5 epimerase have been developed.

### 2.1. Spectrophotometric Assays for the Epimerase Activity of ManC5-Es

Spectrophotometric Assays are usually simple and fast, and have been applied for detecting activities of many enzymatic reactions [27]. The first developed assay for ManC5-Es' activity is the Dische's carbazole reaction [28,29]. In this reaction, the D-mannuronic and L-guluronic acid give different responses to carbazole after being treated with concentrated sulfuric acid [29]. Page et al. [30,31] applied this method to detect epimerase activity in *Azotobacter vinelandii* cultures by calculating the ratio of the spectrophotometer reads at the beginning and end of the reaction. However, this method involves diluting concentrated sulfuric acid in an aqueous solution, which may cause carbonization of polysaccharide if the heat released builds up [32,33], that could cause low reproducibility and low response. Notably, safety should be considered due to the sulfuric acid and carbazole reagents.

Another simple and fast method [34] is to use a combination of specific alginate lyases. The G-specific lyases can cleave the guluronic acid linkages, resulting in an increase in the absorbance at 235 nm due to the generation of 4,5-unsaturated uronic acid (Δ) [34]. Based

on this, a continuous enzyme-coupled assay and fixed-time assay were developed. Wolfram et al. [23] and Gaardlos et al. [35] expressed the GG specific lyase AlyA to degrade samples epimerized by PsAlgG and AvAlgE4, respectively. They then calculated the increased G-content to measure the epimerase activity [23,35]. This assay is more sensitive than the carbazole assay and is suitable for measuring low-level epimerase activity [34]. Thus, it has been widely applied to analyze the epimerization activity of ManC5-Es. However, the epimerase activity of some bifunctional ManC5-Es that contain both epimerase and lyase activity cannot be determined by this method and it requires specific lyase with high activity and stability.

### 2.2. $^1$H-NMR and $^{13}$C-NMR Spectroscopy for C5 Epimerization

The most wildly used methods are the nuclear magnetic resonance (NMR) spectroscopy. According to the X-ray diffraction studies on polyM and polyG, the M residues are in β-linked $^4C_1$ conformation, and the G residues are in α-linked $^1C_4$ conformation, and each unit is independently arranged in alginate solution [6,36]. Knowledge of this principle structure and spin-spin couplings allowed us to study the epimerase activity of ManC5-Es with NMR spectroscopy [37]. Sun et al. [38] used $^1$H NMR to calculate the increasing G-content of alginate after incubation with PmC5A, and determined its epimerase activity. Gaardløs al. [39] combined $^{13}$C NMR and $^1$H NMR to study the epimerase activity and product patterns of AvAlgE7, further understanding the mechanism of AvAlgE7. This method has been widely used to determine both the epimerase and lyase activity (Figure 2) [40].

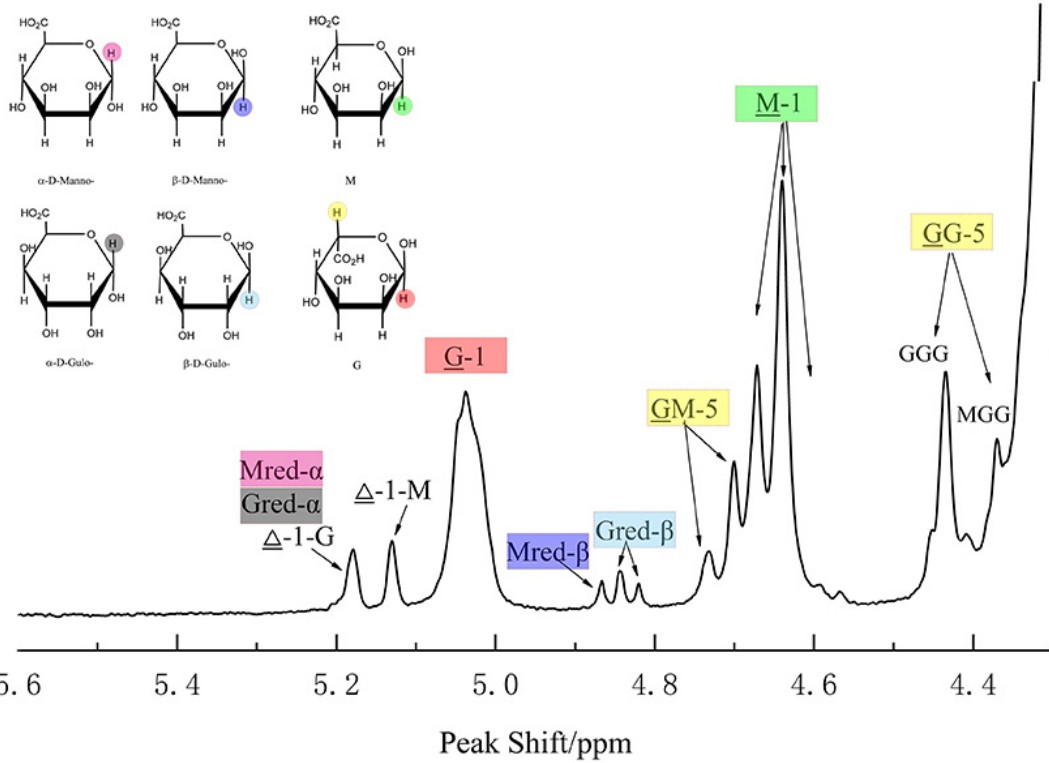

**Figure 2.** Analysis of alginate by $^1$H NMR. The structure of M/G residues and alternative preferred conformations of the anomeric mannosid- and gulosid-uronic acids are named as α-D-Manno -, β-D-Manno-, α-D-Gulo- and β-D-Gulo- in the $^1$H NMR spectrum; The inner residues are named as M and G. Assignments of the H1 and H5 signals are shown in same color with the relative residues; The nonunderlined M or G refers to the neighboring moieties and the numbers refer to the position of the proton in the pyranosyl ring that causes the signal. This figure is prepared from data published previously [38].

Though NMR can provide more detailed information of the epimerized products, some limitations can raise obstacles to the research. Due to the long chain of alginate that would cause low signals at $^1$H NMR spectra prior to record, alginate samples should be partially hydrolyzed to reduce the viscosity for a better resolved resonance [40]. In addition, the spectra is recorded at a high temperature (such as 70 °C, 80 °C, 90 °C, and so on) to further reduce the viscosity and eliminate the water signal from the anomeric region which is responsible for the calculation of the M/G ratio [40]. Several technologies were also used to suppress the water signals and recover target signals that overlap the broad baseline of the intense water resonance, such as presaturation, presaturation followed by a composite pulse, and NOESY-presaturation [40,41]. However, these methods have been proved to cause error in M/G ratio calculation in alginate due to the intensity changes of M-residues that are associated with water [41].

### 2.3. Other Methods for Assaying ManC5-Es Epimerization

In addition to the assays illustrated above, other methods were also developed. The radiometric assay [42], which measures the tritium release in the water when using the [5-$^3$H] alginate as the substrate [39], can also determine the epimerase activity. This is because the epimerization reaction involves an exchange of protons with the solution at the C-5 position of mannuronic acid [43]. However, the lyase reaction also contains the proton exchange that will disrupt the detection of epimerase activity of ManC5-Es [25]. Additionally, the 5-tritiated bacterial alginate is not commercially available, and its preparation is complicated and time-consuming.

Other methods, such as acid hydrolysis [44], $Ca^{2+}$-induced gel precipitation [45], and circular dichroism (CD) [46,47] can also be used to determine epimerase activity. However, the mentioned assays were not widely used due to low response or low accuracy.

In addition to experimental assays, in silico methods such as docking and MD simulations are powerful techniques to study the recognition and binding interactions between ManC5-Es and alginate substrates [35]. However, the QM/MM calculations currently used for the study of the catalytic mechanism of epimerase, such as diaminopimelate epimerase [48] and dipeptide epimerases [49], have not been used for the study of ManC5-Es.

Above all, many methods have been developed to determine the epimerase activity of Manc5-Es based on the difference between M and G, either for qualitative or quantitative analysis. While each method has strengths as well as limitations (Table 1), we suggest that it would be a better choice to combine two or more methods to determine the epimerase activity to further understand the ManC5-Es.

**Table 1.** Methods that applied for the determination of the epimerase activity of ManC5-Es, and their strength and limitations.

| Methods | Strength | Limitations | Reference |
|---|---|---|---|
| Carbazole's reaction | Simple | Low reproducibility, and response | [28] |
| Enzyme-coupled assay | Suitable for measuring low level epimerase activity | Limited by bifunctional activities; Require specific lyase with high activity and stability | [34] |
| NMR | Accurate; More information about products | Require sample pretreatment | [23] |
| Radiometric assay | Sensitive | Require for substrate preparation | [42] |
| Acid hydrolysis | Simple | Low accuracy | [50] |
| $Ca^{2+}$-induced gel precipitation | Simple | Low accuracy | [51] |
| Circular dichroism | Simple | Low response | [52] |

## 3. Characteristics of Mannuronan C-5 Epimerases from Different Sources

Alginate comes from brown algae [4] and some bacteria such as *Azotobacter* and *Pseudomonas* genera [18,53]. Many *ManC5-E* genes have been found in the genome of brown algae [54] and the above two groups of bacterial ManC5-Es have been characterized [17].

### 3.1. Mannuronan C-5 Epimerases from Brown Algae

Algal alginate has more variety in M/G ratios and sequences compared to bacterial alginate [45,54], indicating more potential ManC5-E coding genes in brown algae genome. With the improvement of genomic and transcriptome technology, many putative *ManC5-E* genes has been found. For instance, genomic analysis revealed 105 and 45 candidate *ManC5-E* genes in *Saccharina japonica* and *Ectocarpus siliculosus*, respectively [2,5,55,56]. *De novo* transcriptome analysis suggested that there are 31 potential genes in *Undaria pinnatifida* [57].

Compared with the gene-level analysis, isolation of ManC5-Es from brown algae or heterologous expression of the algal ManC5-Es coding genes is notoriously difficult [58], leading to a rare number of characterized algal ManC5-Es [5,59]. Efforts in heterologous expressing these enzymes have been made continuously and three successful attempts shed light on future studies of the enzyme. Fischl et al. [58] reported the first success in expressing a ManC5-E from brown alga *Ectocarpus siliculosus* in *E. coli* and refolded the enzyme in an active form. Inoue et al. [45] took a different approach to further tackle this problem by expressing the *ManC5-E* gene from *Saccharina japonica* with insect-cell expression systems to achieve the soluble expression. Recently, Zhang et al. [24] successfully expressed the ManC5-E, SjMC5E2 in a soluble form from *S. japonica* with pCold vector and confirmed its epimerase activity *in vitro*. This indicates that refolding the proteins, changing the expression systems or co-expression with chaperone [60], could efficiently help to obtain eukaryotic ManC5-Es.

### 3.2. Characterization of Mannuronan C-5 Epimerases from Azotobacter Species

The research of bacterial epimerases, especially those from *Azotobacter* and *Pseudomonas* species, show significant expression advantages over that of algal ManC5-Es. *Azotobacter vinelandii* is a soil-dwelling bacterium that produces alginate as the major component of cyst, a protective shell formed under harsh conditions [61,62]. The genome of *A. vinelandii* contains two types ManC5-Es coding genes: the $Ca^{2+}$ dependent and independent ManC5-Es [63].

AlgEs, the $Ca^{2+}$ dependent and extracellular enzymes, consist of seven different epimerases AlgE1-AlgE7 [63,64], which are distinguished in terms of their modular substructures. The N-terminus of all AlgEs consist of one or two modules (designated A-module) of 385 amino acids each and an approximately 150-amino-acids long tandemly repeated sequence in the C-terminus (designated R-module) [65,66]. Each R-module contains four to six repeats of the nonameric (nine amino acids) motifs corresponding to the putative $Ca^{2+}$-binding site [63]. Ertesvåg et al. [67] showed that the evolutionary mechanism of these ManC5-Es may be related to duplications, gene fusions, conversions, or unequal recombination.

A-module contains the alginate-binding and catalytic sites and though it can conduct the whole epimerization process alone [67], the R-module is thought to be involved in substrate binding [68]. However, there probably exists a functional dependence between the two kinds of modules because AlgEs have no activity without $Ca^{2+}$ and the presence of R-modules increases the reaction rate, which most likely attributes to R-modules' ability to make $Ca^{2+}$ more available to the catalytic sites [67]. The R-module of AlgE4 shows much more affinity toward alginate than the R-modules of AlgE6 though they present a similar amino acid sequence, and distributions of charged and polar amino acid in the binding site [69]. Random exchanging of partial R-modules of AlgE4 and AlgE2 can generate a new active epimerase with different epimerization patterns [70], indicating the importance of R-modules on the product patterns. The reason for the existence of seven AlgEs is not fully understood. It is possible that AlgEs have the ability to subtly regulate the alginate

produced in the bacterial cells in response to the environmental changes during dormancy, a condition not easily simulated in a lab [71].

Except for the $Ca^{2+}$-dependent AlgEs, *A. vinelandii* also contains a $Ca^{2+}$-independent ManC5-E AvAlgG [71]. Rather than being homologous to AlgEs, AvAlgG exhibits high sequence identity and similar physical organization with PaAlgG, indicating that the biosynthetic machinery has a common evolutionary origin of *P. aeruginosa* and *A. vinelandii* [71]. AvAlgG can conduct the epimerization in the absence of $Ca^{2+}$, but the $Zn^{2+}$ can significantly suppress the epimerase activity. ManC5-Es can be taken advantage of by the *A. vinelandii* to response to environmental changes since the $Ca^{2+}$ has a crucial role in the gel formation of alginate [71].

In addition to *A. vinelandii*, three putative AlgE-like proteins, designed AcAlgE1, AcAlgE2, and AcAlgE3, have been found from *Azotobacter chroococcum* [51]. AcAlgE1 can introduce long GG-blocks into alginate substrate and only displayed epimerases activity. Different from AcAlgE1, both AcAlgE2 and AcAlgE3 are bifunctional with mannuronan C-5 epimerase and alginate lyase activity that is similar with AvAlgE7 [39,51]. However, the reasons for the existence of bifunctional enzymes in *A. chroococcum* are unknown.

*3.3. Characterization of Mannuronan C-5 Epimerases from Pseudomonas Species*

*Pseudomonas aeruginosa* is a mucoid strain which could cause chronic pulmonary infections in patients with cystic fibrosis [72]. The genome of *P. aeruginosa* contains only one periplasmic ManC5-E, PaAlgG [72,73]. PaAlgG is a polymer level alginate C5-mannuronan epimerase [73] but cannot introduce G-blocks [72]. Except for the epimerase activity, PaAlgG can protect the alginate from degradation by the alginate lyase (AlgL) during transport through the periplasm [74]. In addition to *P. aeruginosa*, other *Pseudomonas* species, such as *P. mendocina*, *P. putida*, and *P. fluorescens*, also have *algG* genes [75]. A mannuronan C5-epimerase from *P. mendocina* named PmC5A was reported, and presented both epimerase and lyase activities toward alginate [38]. This is the first report of the periplasmic ManC5-E with bifunctional activities. PmC5A can increase the G-moieties by 9% of alginate and cleave MG and G blocks, indicating the lyase reaction may happen after the epimerization [38]. However, the reasons for the bifunctional activities of PmC5A still remain unknown.

The *P. syringae* genome encodes a $Ca^{2+}$-dependent ManC5-E PsmE that efficiently forms G blocks in vitro [76]. This protein contains one A-module and three R-modules, which share sequence similarity with the corresponding A and R modules of AlgE1-7 from *A. vinelandii*. Similar to AlgEs, the A-module of PsmE is sufficient for epimerization [76]. A hybrid protein containing the A-module from PsmE and the R-module from AlgE4 achieves a 20-fold higher epimerase activity and a final product of G-content up to 96% [76]. Meanwhile, PsmE was shown to epimerize the acetylated alginate with a N-module which encodes an acetyl hydrolase gene [76].

Above all, some ManC5-Es have been characterized, including the prokaryotic and eukaryotic organisms. While most studies are focused on the bacterial ManC5-Es [7], few characteristics of algal ManC5-Es are available [24], leaving much unknown about them. The characteristics of ManC5-Es varies from origins (Table 2), especially for the block distributions that will allow the modifications for desired alginate.

**Table 2.** Main characteristics of the reported ManC5-Es from various origins.

| Name | Orgnism | Optimal Temperature (°C) | Optimal pH | Products | Reference |
|---|---|---|---|---|---|
| | | Ca$^{2+}$-dependent | | | |
| AlgE1 | *A. vineland* | 37 | 6.9 | GG and MG blocks | [66] |
| AlgE2 | *A. vineland* | 55 | 6.5–7.0 | GG blocks | [77] |
| AlgE3 | *A. vineland* | - | - | GG and MG blocks | [63] |
| AlgE4 | *A. vineland* | 37 | 6.7–7.0 | MG blocks | [78] |
| AlgE5 | *A. vineland* | - | - | GG blocks | [63] |
| AlgE6 | *A. vineland* | - | - | GG blocks | [64] |
| AlgE7 | *A. vineland* | - | - | G residues and GG blocks | [64] |
| AcAlgE1 | *A. chroococcum* | - | - | GG blocks | |
| AcAlgE2 | *A. chroococcum* | - | - | GG blocks, Mred and ΔM | [51] |
| AcAlgE3 | *A. chroococcum* | - | - | Mred, Gred, ΔM and ΔG | |
| PsmE | *P. syringae* | 37 | 6.8 | GG blocks | [76] |
| MEP13 | *Ectocarpus* | - | - | GG blocks | [58] |
| | | Ca$^{2+}$-independent | | | |
| PmC5A | *P. mendocina* | 30 | 9 | Δ | [38] |
| SjC5-VI | *S. japonica* | 35 | 7.0–8.2 | - | [45] |
| SjMC5E2 | *S. japonica* | - | - | GG blocks | [24] |

"-" indicates that characteristics have not been determined. "Δ" indicates the 4,5-unsaturated uronic acid. "red" indicates the reducing end.

## 4. Structure and Mechanism of Mannuronan C-5 Epimerases

The mechanism of epimerases reaction is based on the β-elimination reaction of the lyase [79]. In this part, we mainly discuss the structure and important residues that are involved in the catalysis.

### 4.1. The Structure Analysis of Mannuronan C-5 Epimerases

The first solved structure of mannuronic ManC5-E is the A-module of AvAlgE4 which contains a right-handed parallel β-helix fold with an N-terminal α-helix cap and an extended binding groove [80]. The β-helix is composed of four parallel β-sheets, comprising 12 complete turns, and has an amphipathic β-helix near the N terminus [80]. Three extended loops are located over the binding groove, which arrange like the lid loop of some alginate lyase from PL5, -7, and -18 [81]. The conformational change of the lid loop from the open to the closed state results in coverage of the active-site cleft of these alginate lyase [81,82]. It has been shown that the loop extending from the β-helix is essential for the epimerase activity in AvAlgE6-A that would disrupt the substrate binding of the enzyme [83]. The A-module of AvAlgE6 shows 81% amino acid sequence identity with AvAlgE4-A and shapes a right-handed parallel β-helix architecture as well, but has not been further studied.

The structure of R-modules from AvAlgE4 and AvAlgE6 also have been studied (Figure 3) [69,84–87]. The AvAlgE6 R-modules fold into an elongated parallel-roll with a shallow similar to the architecture in AvAlgE4-R (Figure 3) [69]. AvAlgE4-R forms a positively charged patch for arginine and lysine residues placed along the surface of the small groove on the front side of the β-roll, and negatively charged patches for aspartate and glutamate residues located at the turns of the β-roll [84]. Both R-modules of AvAlgE4 and AvAlgE6 consist of one or more RTX motifs (Repeat in ToXin) of the consensus sequence GGXGXDXUXn (Figure 3A) [68,88], and the motif was found in a diverse group of Gram-negative bacteria, presenting the sequence and structure similarity with the metallo-

proteases from *P.aeruginosa* and *Serratia marcescens* [89,90]. This motif stabilizes the fold by binding calcium ions tightly between two neighboring loops in the β-roll [91]. The putative binding sites of the four R-modules in AvAlgE4 have eight basic amino acids and some of the acidic and polar amino acids on the front side of the R-modules are thought to be responsible for dissociating the alginate polymer chain, thus increasing the processivity in AvAlgEs [69,92].

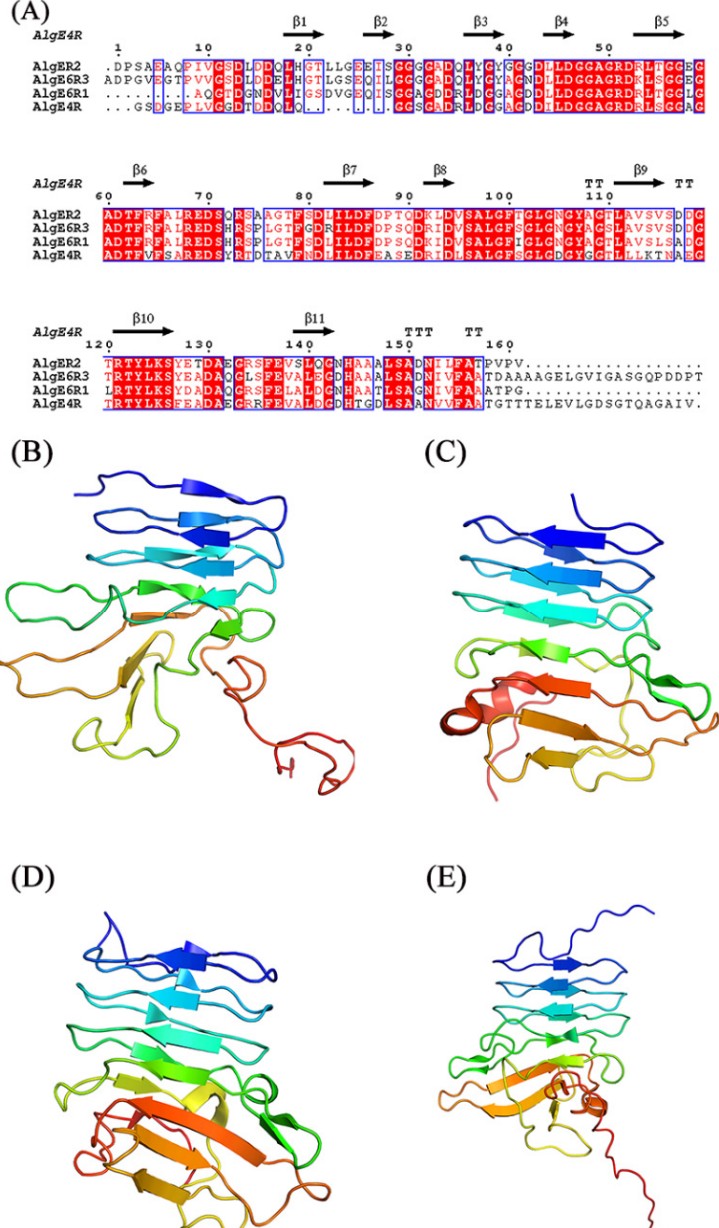

**Figure 3.** Sequence alignment and structures of R-modules of AvAlgE4 and AvAlgE6. (**A**) Alignment of the R-modules of AvAlgE4 and AvAlgE6. The beta strands are represented as arrows and beta turns are marked with TT. The sequence alignment was created using the following sequences from PDB: AvAlgE4R (2AGM), AvAlgE6R1 (2ML1), AvAlgE6R2 (2ML2), and AvAlgE6R3 (2ML3). The figure was created with ESPript 3.0 [93]. (**B**) Overall structure of AvAlgE4R. (**C**) Overall structure of AvAlgE6R1. (**D**) Overall structure of AvAlgE6R2. (**E**) Overall structure of AvAlgE6R3 [69,85,86]. The figures were produced using PyMOL.

Studies have shown that $Ca^{2+}$ binds to a loop between strands β7 and β8 in AvAlgE4 (Figure 4B) [80]. The conformation of the loop is changed when $Ca^{2+}$ is absent. In some lyases with β-helix folding, $Ca^{2+}$ is used to neutralize the carboxylate group of uronic acid during the reaction, which plays a role equivalent to that of $Arg^{345}$ in the *Pseudomonas* periplasmic alginate epimerases PsAlgG [23], illustrating the different mechanisms between them. Given that the AlgEs are extracellular ManC5-Es, the environment would play a key role that affects the mechanism of ManC5-Es.

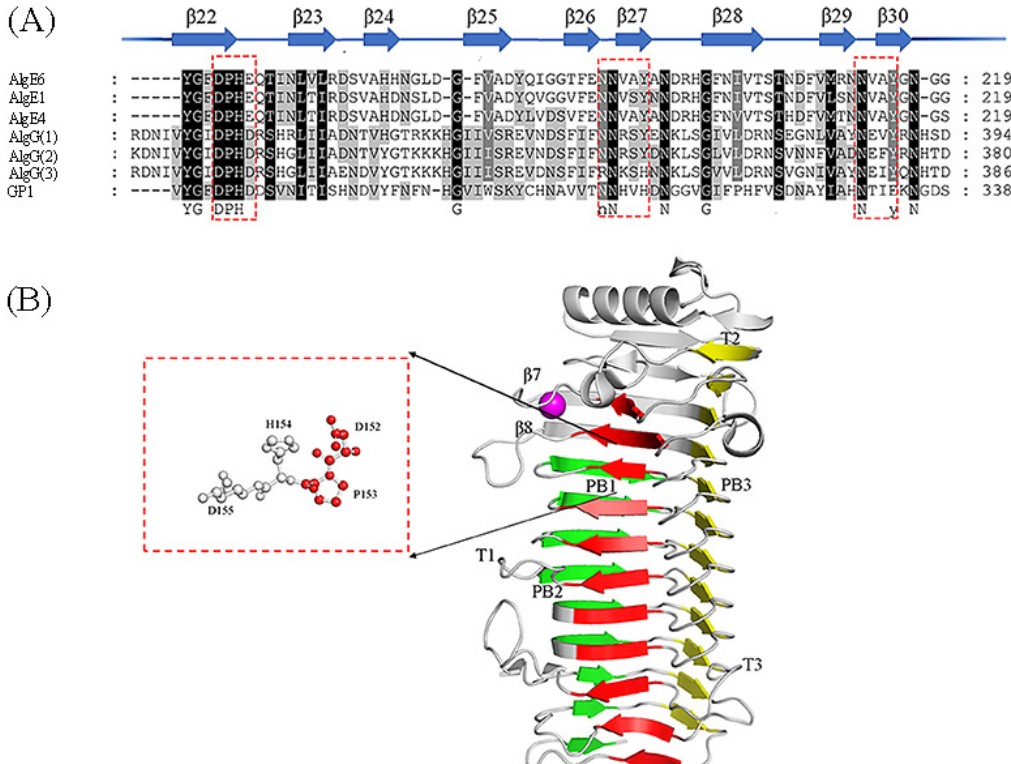

**Figure 4.** Sequence alignment of ManC5-Es with partial sequence and the structure of AvAlgE4A (PDB: 2PYH [80]). (**A**) Conserved amino acid motif among mannuronan C-5 epimerases. The beta strands and relative positions are represented as arrows and numbers respectively. Red boxes show the highly conserved amino acids. This sequence alignment was created using the following sequences from NCBI and UniProtKB website; The extracellular epimerases AvAlgE1 (UniProtKB: Q44494.1), AvAlgE4 (UniProtKB: Q44493.1), and AvAlgE6 (UniProtKB: Q9ZFH0.1) from *A. vinelandii*, and PaAlgG(1) proteins from P. aeruginosa (NCBI: WP_003110465.1), PfAlgG(2) proteins from *P. fluorescens* (UniProtKB: P59828.1) and PsAlgG(3) proteins from *P. syringae* (NCBI: WP_011103483.1), and GP1 proteins (NCBI: CBN80312.1) from *Ectocarpus siliculosis virus*. (**B**) Structure of AvAlgE4A, forming a β-helix fold [80]. The red box shows the catalytic sites. The sheets forming the helix are colored in red (PB1), green (PB2), and yellow (PB3). The T1-T3 turns are also included. The $Ca^{2+}$ ion was in the loop between strands β7 and β8 which could explain the calcium dependence.

Atomic force microscopy revealed that the A-module of AlgE4 binds more strongly to the alginate chain than the complete enzyme, indicating that the R-module is involved in the regulation of the enzyme-substrate binding strength [66]. Nevertheless, the AvAlgE4-R on its own is also able to bind to the alginate chain [84] and none of the individual R-modules of AvAlgE6 showed the obvious affinity to alginate like AvAlgE4-R, though the amino acid sequence and structures are very similar (Figure 5). There is no report about the complete structure of multimodular ManC5-Es up to now. The low resolution SAXS-measurements of AvAlgE4 and AvAlgE6 exhibit an elongated shape with some flexibility between the individual modules [69]. A combination of the R-module from AvAlgE4 and the A-module

from AvAlgE6 resulted in a novel epimerase called AvAlgE64 displaying better G-block forming abilities than AvAlgE6 [69], indicating that the substrate binding event can alter the products patterns.

**Figure 5.** The mechanism of ManC5-Es and alginate lyase. (**A**) The function of mannuronan C-5 epimerase. (**B**) Typical proposed unified mechanism for alginate lyases (**left**) and epimerases (**right**) [79]. AA1-AA3 represents the amino acid residues on the enzymes and are shown in color blocks. The hydrogen atoms involved in the epimerase reaction are also shown in color blocks.

Another structure of a non-module mannuronic epimerase, PsAlgG from *P. syringae*, was solved in 2014 (Figure 5B) [23]. PsAlgG adapts a long right-handed parallel β-helix with an elaborate lid structure. This fold is common to many carbohydrate hydrolases and lyases according to the Carbohydrate-Active enZYmes (CAZy) database, including the alginate lyase from PL6 and -31 [94]. Studies showed that PsAlgG contains nine 24-mer repeats at its C-terminus [23], which is predicted to be carbohydrate-binding and sugar hydrolase (CASH) domains. The CASH domain which are commonly found in carbohydrate lyases forms parallel right-handed β-helices (RHβHs) that folds and forms a spring-like structure with a shallow groove on one face which accommodates long-chain linear polysaccharides [95,96]. Moreover, it has been shown that ManC5-Es from *Laminaria digitate* and A-module of AvAlgEs from *A. vinelandii* also have a secondary structure similar to the PsAlgG [54]. Therefore, this kind of structure appears to be conserved among the ManC5-Es.

Overall, the reported structures of ManC5-E are relatively rare. Especially, there is no structure of MansC5-Es from the algal origin. This could be due to the obstacles to express the ManC5-Es from algae. Both the $Ca^{2+}$-dependent and -independent ManC5-Es adopt the right-handed parallel β-helix fold [23], indicating that the architecture is essential for

the epimerization of ManC5-Es, which allows either the long chain alginate binding or processing in the catalytic cleft.

### 4.2. The Catalytic Mechanism of Mannuronan C-5 Epimerases

The proposed mechanism of both lyase and epimerase reaction has been described (Figure 5): (1) neutralization of the carboxylate group; (2) abstraction of the proton of the C-5; (3) either a β-elimination of the 4-O-glycosidic bond (the lyase reaction) or the replacement of the proton of C-5 (the epimerase reaction) [79]. According to the model purposed by Gacesa [79], the catalytic triad amino acids AA1, AA2, and AA3 are different in periplasmic and extracellular epimerases (Figure 5B). Additionally, another proposal of the mechanism is that the epimerization proceeds via β-elimination with the glycosidic linkage breaking, followed by reformation of the glycosidic bond with protonation on the opposite face of the residue undergoing epimerization [97], which resembles dermatan epimerization. But this mechanism of ManC5-Es was not yet fully recognized.

Structural analysis showed that the conserved motif DPHD of AlgGs and DPHE of A-modules (Figure 4A), lying in the center of grooved face, are important for catalytic function [23]. Tyr functions as a general base (AA2 in Gacesa's model) while His acts as a general acid (AA3 in Gacesa's model) [79]. The carboxylic acid moiety of the bound mannuronic acid points to the interior of the enzyme, establishing hydrogen bonds with the carboxylic acid side chains of two Asp, and the acidic micro-environment ensures that the sugar carboxylate is protonated (AA1 substrate stabilization) [23].

In addition, the motif of NNRSY and NLVAY is also conserved among all ManC5-Es (Figure 4A) [23]. The mutation of N362A and N367A may cause the proteins misfolding, whereas mutations of S364A and Y365F have no effect on the enzyme's activity in vitro [23]. Moreover, the motif NNRSY is common in both epimerases and lyases by different versions, indicating its importance for catalysis or binding [23]. All of these indicated that the lyases and the polymer-level epimerases have essentially a common mechanism of action [23,25]. In terms of the two similar mechanisms, Rozeboom et al. [80] proposed some explanations for how epimerase prevents the lyase reaction. One way is that the interaction between Tyr and Arg may effectively remove the proton from the active site; another way is that the hydrogen bonding interaction between the O-3 hydroxyl group of the +1 sugar and the ring O-5 atom of the -1-sugar residue may help to prevent the two sugars from moving apart [80].

Based on the structure of AvAlgE4-A, the charged residues surrounding the binding groove that form the hydrogen-bonding network have been studied [35]. Molecular dynamics and binding free energy analyses show that, in the binding groove, positively charged residues contributed favorably and negatively charged residues contributed unfavorably to the stability of the analyzed binding poses, and a stronger binding action toward the substrate than the products [35]. The processive action of AvAlgE4 is described in three steps: (1) initial binding to the substrate, participated by the positively charged residues Arg[195], Arg[249], Arg[276] and Arg[343] at the C-terminal; (2) processive movement aided by residues Lys[84], Arg[90], and Lys[117] at the N-terminal; and (3) dissociation from the substrate involved charge pair Arg[342] and Asp[345], Asp[173] and Asp[345] complicated rearrangements of the H-bonding network and electrostatic environment of the binding groove [35]. Several hybrid epimerase AvAlgE64s with partial region from the AvAlgE6 A-module and AvAlgE4 was generated, and further study identified the loop protruding out from the 10th turn in the β-Helix as important for determining the epimerization pattern. Specifically, Tyr[307] in the loop can influence the epimerization activity and product file [98].

Among all the reported ManC5-Es, several members of them present both alginate lyase and epimerase activity [38,39,51]. It has been proposed that the reaction mechanism of these enzyme activities are very similar [79]. Later, it was proved that a common site is responsible for both activities [25]. The hybrid protein AvAlgE7-E1 containing the N-terminal part of AvAlgE7 and the C-terminal part of AvAlgE1 exhibited dual activities, indicating the importance of N-terminal of AvAlgE7 for the lyase activity [25]. Most recently,

mutagenesis experiment of AvAlgE7 carries out that Arg[148] displays a significant effect on alginate lyase activity by influencing the catalytic acid Tyr[149] proton donation toward either the other side of the sugar ring (the epimerase reaction) or the glycoside bond (the lyase reaction) [39]. And residues Glu[117], Arg[148], and Lys[172] are thought to be responsible for positioning the substrate by electrostatic interactions and group pKa perturbing [39].

In summary, though the reported structures of ManC5-Es are relatively rare, the well-studied enzymes structure and catalytic mechanism deepen our understanding of the reaction process of ManC5-Es in vitro. Though Manc5-Es come from various sources exhibiting diverse characteristics, they share the same structure of β-helix (RHβH) architecture, and conserved motifs, NNRSY, NLVAY, and DPHD(E), indicating the importance of the similarities for catalysis.

### 5. Conclusions and Prospects

Although the research on ManC5-Es has made breakthroughs in recent years, there are still many unknowns. Firstly, all the determined structures and proposed catalytic mechanisms of ManC5-Es are from bacteria; yet the knowledge of the mechanisms involved in brown algae is still preliminary, partly because it is still challenging to heterogeneously express the genes. Therefore, solving this problem will help further analyze the function of ManC5-E in the alginate biosynthesis as well as the cell-wall metabolism in brown algae. Several strategies have shed light of the expression of eukaryotic ManC5-Es, including refolding the insoluble proteins [58], changing the expression systems [45], and co-expression with chaperones [24]. This would allow us to further understand the characteristics and mechanism of eukaryotic ManC5-Es.

Secondly, $Ca^{2+}$ is essential for the epimerase activity of the AlgEs, and other multimodular $Ca^{2+}$-dependent ManC5-Es. Though it has been reported that the R-modules participate in modulating the substrate binding and product pattern through regulating the $Ca^{2+}$, the detailed mechanisms are still unknown. Thus, it is important to further clarify the roles of $Ca^{2+}$ in the catalysis of $Ca^{2+}$-dependent ManC5-Es based on the structure of ManC5-Es.

Thirdly, though the characteristics of AlgEs have been extensively studied, the mechanism inside the substrate recognition and binding, and product patterns are still unclear since they share high homology but exhibit different substrate preferences and product patterns. Thus, additional studies will focus on the underlying molecular mechanism that control the processive movement of the ManC5-Es. Consequently, the answers to these questions would lead to a deeper understanding of the catalytic mechanism of ManC5-Es and further help to apply ManC5-Es in industries, achieving the production of the desired types of alginates.

**Author Contributions:** Z.X.: Investigation, Writing—Original Draft, M.S.: Investigation, Writing—Original Draft. T.L.: Writing—Review and Editing. M.Z.: Writing—Review and Editing. H.Y.: Supervision, Writing—Review and Editing, Funding acquisition. All authors have read and agreed to the published version of the manuscript.

**Funding:** This work was supported by Dalian Science and Technology Innovation Fund-Key & Major Subject (2020JJ25CY017); Liaoning Provincial Marine Economic Development Project (2022-47); Outstanding Member Fund of CAS Youth Innovation Promotion Association (Y201939).

**Data Availability Statement:** Not applicable.

**Conflicts of Interest:** The authors declared no conflict of interest.

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
