# Peer review of "Mannuronan C-5 Epimerases: Review of Activity Assays, Enzyme Characteristics, Structure, and Mechanism"

_catalysts, doi:10.3390/catal13010028_

Round 1
Reviewer 1 Report
In the manuscript of Xiao et al. the authors summarise recent findings regarding the methods applied to study ManC5-Es reaction, the description of Man5C-Es from different organisms as well as the structure and mechanism of ManC5-Es. The manuscript is well organised and well written. I would like to propose the following minor suggestions that could improve the quality of this manuscript.
1. The authors proposed that the M/G ratios as well as other parameters (such as sequence, molecular weigth and block distributions) define the physical properties of alignate. I think it would benefit to also include the ranges of these ratios (or perhaps mention the most common) and the sizes of alignate.
2. In the section regarding the structure and mechanism of ManC5-Es authors mention the application of computational methods in description of catalytic mechanism of ManC5-Es (section 4.2). I would like to ask if these methods can be used to analyse ManC5-Es reaction? If so, I think it would be helpful to add short comment in section 2.
3. In section 4.1 (page 7) authors mention that the β-helix of AlgE4-A is essential for epimerase activity. It would be beneficial to add a short comment explaining the reason of its significance.
4. Some abbreviations might need explanation (for example. CAZys).
Appart from these following suggestions/questions I do not have any critical comments.
Author Response
Response to Reviewer1 comments:
In the manuscript of Xiao et al. the authors summarise recent findings regarding the methods applied to study ManC5-Es reaction, the description of Man5C-Es from different organisms as well as the structure and mechanism of ManC5-Es. The manuscript is well organised and well written. I would like to propose the following minor suggestions that could improve the quality of this manuscript.
Many thanks to you for your constructive comments on our manuscript. We have revised our manuscript accordingly, and all these changes improved the quality of our manuscript. Please see below and find our answers to your questions point by point:
- The authors proposed that the M/G ratios as well as other parameters (such as sequence, molecular weight and block distributions) define the physical properties of alginate. I think it would benefit to also include the ranges of these ratios (or perhaps mention the most common) and the sizes of alginate.
Response: Thank you for your comment. Indeed, the M/G ratio is an important property of alginate. So far, about 200 different kinds of alginates have been identified and extracted from natural resources. The M/G ratio of alginate varies significantly among source organisms. The ratio of typical brown seaweed species is mainly between 0.43 and 2.52 (Saji et al. Sustainability, 2022, 14 (9): 5181). As for the sizes of alginate, the molecular weight of commercially available sodium alginates typically ranges between 32,000 and 400,000 g/mol (Lee et al. Progress in Polymer Science, 2012, 37: 106-26). The additional information has been added accordingly in section 1 (lines 34-36).
- In the section regarding the structure and mechanism of ManC5-Es authors mention the application of computational methods in description of catalytic mechanism of ManC5-Es (section 4.2). I would like to ask if these methods can be used to analyse ManC5-Es reaction? If so, I think it would be helpful to add short comment in section 2.
Response: Thank you for your concerns and helpful suggestion. Docking and MD simulations are suitable methods to study the recognition and binding interactions between enzymes and substrates, and these methods have been used in the study of AlgE4 (Gaardløs et al. Glycobiology. 2021, 31: 1616-35). In terms of in silico analysis of enzyme reaction mechanism, QM/MM calculation is more useful, and has been used in the study of the catalytic mechanism of epimerases such as diaminopimelate epimerase (Stenta et al. J Chem Theory Comput. 2009, 5: 1915–30) and dipeptide epimerases (Tripathi et al. Chemistry–An Asian Journal. 2022, 17: e202200528). However, so far, the QM/MM method has not been used for the study of ManC5-Es. A short description has been added in section 2.3 of the main text (lines 135-139).
- In section 4.1 (page 7) authors mention that the β-helix of AlgE4-A is essential for epimerase activity. It would be beneficial to add a short comment explaining the reason of its significance.
Response: Thank you for your helpful advice. According to the study conducted by Dypas, the structures at the C-terminal end of AlgE6A (including the β-helix region) are more important for epimerase functionality than first thought. Removal of the C-terminal segment containing the β-helix region led to completely lost activity. The reason for the especially important role of this helix would be its involvement in substrate binding (Dypås LB. Norwegian University of Science and Technology; 2015ï¼›Master thesis). A short comment has been added in section 4.1 accordingly (lines 260-261).
- Some abbreviations might need explanation (for example. CAZys).
Response: Thank you for your careful check and helpful advice. CAZy stands for the Carbohydrate-Active EnZymes database and the manuscript has been revised accordingly in section 4.1 (line 315).
Appart from these following suggestions/questions I do not have any critical comments

Reviewer 2 Report
The manuscript entitled “Mannuronan C-5 Epimerases: Review of Activity Assays, Enzyme Characteristics, Structure, and Mechanism” is a literature revision of epimerases, that gathers recent information related to enzyme activity assay and enzyme characteristics. This enzyme is responsible for alginate modifications that can result in several applications.
The manuscript is clear, well written, and important for researches in this field.
Specific comments:
Abstract: Consider revising the abstract. Now it is like the summary of the introduction. A revision paper has also results and they should be present here. The findings of the revision should be more evident. Which are the most used methods for enzyme activity? What are the characteristics of the enzyme that most articles reveal? .
“Mannuronan C-5 epimerase (ManC5-E) is present in brown algae and certain bacteria, 10 such as Azotobacter and some Pseudomonas species.” It is present or it is produced by?
Introduction section
Lines 43-46. Revise sentence. It is confusing.
“The biosynthesis pathway of alginate in bacteria involves a group of genes [15], among which, the mannuronan C5-epimerase...” C5-epimerase is an enzyme, not a gene...
“All alginate producing organisms harbor ManC5-Es genes [19] and some of them from bacteria and brown algae have been isolated and characterized.” The genes or the enzyme? The end of the sentence should have a reference or references.
Lines 51-52: revise the sentence.
Line 51: The authors say that systematic reviews are needed on this subject. But the present paper is not a systematic review.
Line 64: …and THEIR mechanism
Line 73: …after BEING treated…
Line 75: What is “reading ratio”. This sentence is confusing.
“However, this method requires critical operations and conditions, such as the temperature and water...” Which temperature? And what do you mean by “water”. Water content, water activity, ...? why they are considered critical?
Some citations are missing the reference number. For example, lines 74 and 84. After the name of the author, the reference number should appear. Check it all over the manuscript.
Lines 84-87: revise the sentence.
Table 1 and Figure 2 are not mentioned in the text. They must be.
Section 3.2: use “Azotobacter” underlined in the title. Same for section 3.3.
Table 2: symbols and abbreviations must be specified in footnote.
Author Response
Response to Reviewer 2 Comments:
The manuscript entitled “Mannuronan C-5 Epimerases: Review of Activity Assays, Enzyme Characteristics, Structure, and Mechanism” is a literature revision of epimerases, that gathers recent information related to enzyme activity assay and enzyme characteristics. This enzyme is responsible for alginate modifications that can result in several applications.
The manuscript is clear, well written, and important for researches in this field.
Thank you for your constructive suggestions for our manuscript. We have revised our manuscript according to your suggestions. All these changes improved the quality of our manuscript.
Specific comments:
Abstract: Consider revising the abstract. Now it is like the summary of the introduction. A revision paper has also results and they should be present here. The findings of the revision should be more evident. Which are the most used methods for enzyme activity? What are the characteristics of the enzyme that most articles reveal?
Response: Thank you for your helpful advice. Currently, the most used method for ManC5-Es activity assay is nuclear magnetic resonance spectroscopy, and the properties of the enzyme that most articles revealed are the product distributions in terms of the structure of ManC5-Es. The abstract has been carefully revised accordingly with the text “Mannuronan C-5 epimerases (ManC5-Es) are produced by brown algae and some bacteria, such as Azotobacter and some Pseudomonas species. It can convert the transformation of β-D-mannuronic acid (M) to α-L-guluronic acid (G) in alginate with different patterns of epimerization. Alginate with different compositions and monomer sequences possess different properties and functions, which have been utilized in industries for various purposes. Therefore, ManC5-Es are key enzymes that are involved in the modifications of alginate for fuel, chemical, and industrial appli-cations. Focusing on ManC5-Es, this review introduces and summarizes the methods of ManC5-Es activity assay especially the most widely used nuclear magnetic resonance spectroscopy method; characterization of the ManC5-Es from different origins especially the research progress of its enzymatic properties and product block distributions; the catalytic mechanism of ManC5-E based on the resolved enzyme structures. Besides, some potential future research directions are also outlooked.” in lines 10-20.
“Mannuronan C-5 epimerase (ManC5-E) is present in brown algae and certain bacteria, 10 such as Azotobacter and some Pseudomonas species.” It is present or it is produced by?
Response: Thank you for your concerns. We are sorry for our unclear statement. The ManC5-Es are produced by the brown algae and certain bacteria. We have revised “present” into “produced by” in the abstract (line 10).
Introduction section
Lines 43-46. Revise sentence. It is confusing.
“The biosynthesis pathway of alginate in bacteria involves a group of genes [15], among which, the mannuronan C5-epimerase...” C5-epimerase is an enzyme, not a gene...
Response: Thank you for your helpful advice. We have revised this sentence in the main text with “The biosynthesis pathway of alginate in bacteria involves a group of enzymes [16].The mannuronan C5-epimerase (ManC5-E), that can convert M into G at the polymer level, can significantly alter the properties of alginate [19].” in section 1 (lines 44-45).
“All alginate producing organisms harbor ManC5-Es genes [19] and some of them from bacteria and brown algae have been isolated and characterized.” The genes or the enzyme? The end of the sentence should have a reference or references.
Response: Thank you for your helpful advice. The sentence in the manuscript has been revised to read “All alginate producing organisms harbor ManC5-Es genes [7] and some of which from bacteria and brown algae have been subcloned and the encoded enzymes characterized [23, 24].” in section 1 (line 45-47).
Lines 51-52: revise the sentence.
Response: Thank you for your helpful advice. We have revised this sentence into “With the development regarding the characteristics and mechanisms of ManC5-Es, it requires a timely review to summarize the progress of ManC5-Es in recent years.” in section 1 (lines 50-51).
Line 51: The authors say that systematic reviews are needed on this subject. But the present paper is not a systematic review.
Response: Thank you for your helpful advice. Indeed, this review is focusing on the timely progress of the methods of epimerase activity assay, major characteristics of algal and bacterial ManC5-Es, and the catalytic mechanism of ManC5-E based on the resolved enzyme structures. It is not a systematic review regarding the full properties of ManC5-Es. Therefore, we have modified our statement accordingly in section 1 (line 55).
Line 64: …and THEIR mechanism
Response: Thank you for your careful check. We have revised “and mechanism of them” into “their mechanism” in section 1 (line 58).
Line 73: …after BEING treated…
Response: Thank you for your careful check. We have revised “after treated” into “after being treated” in section 2.1 (line 77).
Line 75: What is “reading ratio”. This sentence is confusing.
Response: Thank you for your comment. The sentence in the manuscript has been revised to read “…applied this method to detect epimerase activity in Azotobacter vinelandii cultures by calculating the ratio of the spectrophotometer reads at the beginning and end of the reaction.” in section 2.1 (line 79).
“However, this method requires critical operations and conditions, such as the temperature and water...” Which temperature? And what do you mean by “water”. Water content, water activity, ...? why they are considered critical?
Response: Thank you for your comment. We were originally to illustrate that the reaction temperature and aqueous solution can affect the Dische’s carbazole reaction. We are sorry for our mistaken statement. The Dische’s carbazole reaction involves diluting concentrated sulfuric acid in an aqueous solution, which may cause carbonization of polysaccharide if the heat released builds up. We have modified our statement in the manuscript accordingly in section 2.1 (line 77).
Some citations are missing the reference number. For example, lines 74 and 84. After the name of the author, the reference number should appear. Check it all over the manuscript.
Response: Thank you for your careful checks and helpful advice. We have carefully checked and corrected our manuscript.
Lines 84-87: revise the sentence.
Response: Thank you for your helpful advice. We have revised this sentence into “Wolfram et al. [23] and Gaardlos et al. [35] expressed the GG specific lyase AlyA to degrade samples epimerized by PsAlgG and AvAlgE4, respectively. They then calculated the increased G-content to measure the epimerase activity.” in section 2.1 (line 89-91).
Table 1 and Figure 2 are not mentioned in the text. They must be.
Response: Thank you for your careful check. Sorry for our mistakes. The main text cites Table 1 (section 2.3) and Figure 2 (section 2.2) in lines 142 and 102, respectively.
Section 3.2: use “Azotobacter” underlined in the title. Same for section 3.3.
Response: Thank you for your careful check and helpful advice. The underline marks have been added to “Azotobacter” and “Pseudomonas” accordingly in section 3.2 (line 177) and 3.3 line (223), respectively.
Table 2: symbols and abbreviations must be specified in footnote.
Response: Thank you for your careful check and helpful advice. We have added the symbols and abbreviations accordingly under table 2 (line 251-252).
